# Learning to Unlearn: Machine Unlearning via Learning the Unlearning Behaviors

## Abstract

Various machine unlearning techniques have been developed in response to privacy legislation requirements. These techniques enable individuals to exercise their legal right to have their data $D_f$ removed from a machine learning model. This process, commonly referred to as machine unlearning, is accomplished via the use of an unlearning function denoted as $U$. Existing methods design an intricate $U$ to unlearn $D_f \subset D$ from a previous model $A(D)$ learned on $D$, so that the unlearned model performs as closely as possible to the retrained model $A(D \setminus D_f)$. However, these methods often take a long time due to the complex structures of $U$. Inspired by Learning to Optimize, in this paper, we introduce the first learning-based model-agnostic approach named Learning-to-UnLearn (or L2UL) based on a distribution perspective, which acquires a simple $U$ via learning. Our experimental results demonstrate that the accuracy achieved by L2UL is comparable to that of retraining, while also exhibiting impressive efficiency.

## 1 Introduction

With the progress of society, there has been a significant increase in the amount of available data. This growth has enriched our lives by allowing us to access a wide range of images and videos shared by numerous individuals online. However, it also presents risks to the privacy of users. In response to this, legislative measures like CCPA (California Consumer Privacy Act), PIPEDA (Personal Information Protection and Electronic Documents Act), and GDPR (General Data Protection Regulation) have been introduced to establish rules that protect users' privacy by giving users the right to delete their data.

The task of erasing data from a machine learning model is named machine unlearning Cao & Yang (2015); Bourtoule et al. (2021); Nguyen et al. (2022b).

A naive approach is to retrain the model on the remaining data, but this is expensive. Proposing efficient and effective machine unlearning methods has attracted the attention of researchers in recent years.

Current approaches focus on designing a complex unlearning function $U$ so that the unlearned model performs close to the retrained model. For example, some methods design $U$ as a retraining on all or part of the data, which is very time-consuming and requires retraining every time on an unlearning request. Some methods necessitate the calculation of the Hessian matrix for the designed $U$, which further increases the computational cost.

The goal of an unlearning function $U$ is to produce a model that has unlearned the user data and performs close to the model retrained on the remaining data. ***Why don't we learning to unlearn directly?***

Inspired by Learning to Optimize (L2O) Chen et al. (2022); Tang & Yao (2024), traditional optimization algorithms like Adam are manually designed by human experts through theoretical derivation. In contrast, the L2O approach aims to let machines automatically learn how to optimize. Specifically, the L2O model takes the current state of the optimization iteration as input (such as the current solution and its gradient) and learns to predict the update amount for the next iteration. *Instead of designing a complex and highly time-consuming $U$, we use the retrained model as ground truth to learn a $U$ whose structure is simple but enough to unlearn $D_f$ effectively.* In this way, we can achieve

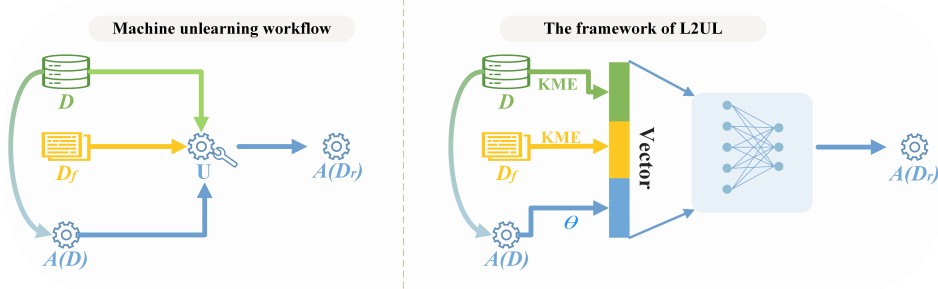

Figure 1: The Machine unlearning workflow and the framework of our proposed method L2UL. The model $A(D)$ is learned on dataset $D$ (which includes $D_f$) by $A$. An unlearning algorithm $U$ unlearns $D_f$ from $A(D)$ to obtain a revised model, which is expected to be equal to the retrained model $A(D \setminus D_f)$.

the goal of unlearning much more efficiently. Although many methods claim to learning to unlearn Cha et al. (2024); Ma et al. (2022a), they usually refer to learning a new decision boundary to achieve forgetting rather than learning the behavior of unlearning.

In this paper, we introduce a model-agnostic framework called Learning-to-UnLearn (L2UL), which employs a machine-learning approach to learn a $U$. The main contributions of this work are:

1. Pioneering a new framework of machine unlearning via learning, named Learn-to-UnLearn (L2UL).

2. Providing the generalization bound of a model unlearned by L2UL in the context of Logistic Regression and Multi-Layer Perceptron (MLP).

3. Conducting an empirical validation of the effectiveness and efficiency of our proposed method.

The advantages of the proposed L2UL over current methods are two-fold:

(i) Unlike existing methods, in L2UL, the acquisition of $U$ is achieved through learning rather than designing.

(ii) Due to the simple structure of $U$, the execution time of unlearning via the learned $U$ is very short.

## 2 PROBLEM FORMULATION

Let $\mathcal{X}$ denote the sample space, and $\mathcal{Y}$ denote the label space. A data point is an element in $\mathcal{Z} = \mathcal{X} \times \mathcal{Y}$. A hypothesis function $h : \mathcal{X} \to \mathcal{Y}$ is learned on training set $D \subset \mathcal{Z}$ by algorithm $A$ to assign $y \in \mathcal{Y}$ to $x \in \mathcal{X}$.

**Definition 2.1.** A machine learning algorithm is a map $A$ from a subset of $\mathcal{Z}$ to a hypothesis function $h \in \mathcal{H}$, where $\mathcal{H}$ is the space of all hypothesis functions of the algorithm.

After $A$ learned a model from a dataset $D$, $A(D)$ can be applied to a test set. When some users request to have their data $D_f \subset D$ deleted, model $A(D)$ is obliged to have unlearned their data. The task of unlearning data from a learned model is machine unlearning.

Figure 1 shows the workflow of machine unlearning. To fulfill a request to unlearn $D_f$ from $A(D)$, an unlearning algorithm $U(D, D_f, A(D))$ produces a revised model which is expected to be the same or close to the model $A(D \setminus D_f)$ retrained on dataset $D \setminus D_f$.

**Definition 2.2.** A machine unlearning algorithm is a map $U : 2^{\mathcal{Z}} \times 2^{\mathcal{Z}} \times \mathcal{H} \to \mathcal{H}$, where $2^{\mathcal{Z}}$ is the power set of $Z$, which is set containing all of $\mathcal{Z}$'s subset. $U$ takes $D, D_f, A(D)$ as input, and outputs the unlearned model, which is expected to be as close to the retrained model $A(D \setminus D_f)$ as possible.

**Definition 2.3.** Exact Unlearning: given a randomized learning algorithm $A$, a dataset $D$, and a data subset $D_f \subset D$ to be forgotten, an unlearning algorithm $U$ is an exact unlearning process

iff $\mathcal{P}(U(D, D_f, A(D))) = \mathcal{P}(A(D \setminus D_f))$, where $\mathcal{P}(A(D))$ defines the distribution of all models trained on a dataset $D$ by a learning algorithm $A$.

The naive unlearning algorithm simply just retrains a new model on $D \setminus D_f$ to achieve exact unlearning. However, it is expensive to retrain when $|D \setminus D_f|$ is large.

## 3 RELATED WORK

Many machine unlearning algorithms that remove some user data from a previously trained model without retraining have been proposed recently. According to the workflow shown in Figure 1, we divide the related work into three categories: $D$-oriented, $D_f$-oriented, and $A(D)$-oriented unlearning.

$D$-**oriented unlearning** operates on subsets of $D$ such that only retaining on some subsets is required upon a request to unlearn $D_f$. For example, SISA Bourtoule et al. (2021) divides the training data $D$ into shards and divides each shard into slices. Each shard is used to train a constituent model, incrementally incorporating slices and preserving its parameters until the training set is extended with a new slice. When $D_f$ is required to be forgotten, retraining is limited to the specific constituent model whose shards encompass $D_f$. Many methods share a similar intuition, such as Cao & Yang (2015); Brophy & Lowd (2021); Lin et al. (2023); Ginart et al. (2019); Gupta et al. (2021); Wu et al. (2023).

$D_f$-**oriented unlearning** considers the impact of forgotten data on the model, through influence functions Wu et al. (2022); Guo et al. (2020); Warnecke et al. (2023) and Markov Chain Monte Carlo-based sampling Nguyen et al. (2022a); Fu et al. (2021). When $D_f$ needs to be forgotten, the impact of $D_f$ on the model can be erased accordingly.

$A(D)$-**oriented unlearning** considers how the parameters of a learned model should be updated after $D_f$ is forgotten from the training set Sekhari et al. (2021); Neel et al. (2021); Ma et al. (2022b); Tarun et al. (2023b); Wu et al. (2020). Tarun et al. (2023a) updates the parameters by adding noise to the parameters to forget and then tuning the model on the remaining data $D \setminus D_f$. Few-shot learning Chundawat et al. (2023b) is used in the case where $D$ is unavailable, and only model $A(D)$ is known.

Although existing works can produce an unlearned model close to the retrained model, none of them use a learning technique to obtain $U$. Our proposed L2UL, an $A(D)$-oriented unlearning algorithm, tries to learn the unlearning function $U$, instead of designing one like current works.

## 4 PROPOSED METHOD

### THE FRAMEWORK OF L2UL

Recall Definition 2.2 and Figure 1, $U$ has three inputs, $D$ (represented as a $|D| \times d$ matrix), $D_f$ (represented as a $|D_f| \times d$ matrix) and the parameter vector $\theta$ of model $A(D)$. A machine unlearning algorithm $U$ must satisfy the following properties:

**Proposition 4.1.** *The output of $U$ should remain unchanged after any two rows in the data matrix of $D$ or $D_f$ are exchanged.*

This is easy to understand because exchanging any two rows of $D$ and $D_f$ does not change the dataset.

An effective method that satisfies the Proposition 4.1 is to represent $D$ and $D_f$ as distribution $\mathcal{P}_D$ and $\mathcal{P}_{D_f}$. No matter how the order of points of the dataset changes, its distribution remains the same.

Therefore, we redefine Definition 2.2 as:

**Definition 4.2.** A machine unlearning algorithm is a map $U: \mathscr{P} \times \mathscr{P} \times \mathcal{H} \to \mathcal{H}$, where $\mathscr{P}$ is the space containing all the probability distributions.

Different from Definition 2.2, Definition 4.2 defines machine unlearning from the perspective of distribution. Based on Definition 4.2 and the unlearning workflow shown in Figure 1, L2UL is naturally proposed by directly using three components of the unlearning workflow: $D, D_f$, and

$A(D)$. Figure 1 shows the framework of L2UL, which uses a neural network $N_U$ [1] to produce the unlearned model $A_{\theta'}$.

In order to vectorize distribution $\mathcal{P}_D$ and $\mathcal{P}_{D_f}$ as input to $N_U$, we employ Kernel Mean Embedding (KME) Smola et al. (2007) to transform distribution $\mathcal{P}_D$ and $\mathcal{P}_{D_f}$ to vector $\mu_D$ and $\mu_f$ in Reproducing Kernel Hilbert Space (RKHS).

Kernel mean embedding is a nonparametric method which uses an element of an RKHS as the representation of a probability distribution. The mean feature map $\mu_{\mathbb{P}} = \mathbb{E}[\kappa(\cdot, D)]$ of points $X$ based on kernel $\kappa$ embeds distribution $\mathbb{P}$ into RKHS, and preserve all of the statistical features of $\mathbb{P}$.

In our framework, when using some common kernel such as Gaussian kernel, Laplacian kernel, etc. $\mu_{\mathbb{P}}$ will map $\mathbb{P}$ into a infinite-dimensional space, which cannot be used as input to the neural network. In other words, we need a kernel whose RKHS dimension is finite.

One approach is to use low-rank representations of kernel matrix. The most popular examples are Nyström method Kumar et al. (2012); Williams & Seeger (2000); Musco & Musco (2017) and the Random Fourier Features Rahimi & Recht (2007); Yang et al. (2012). However, these are all approximate methods and will reduce effectiveness.

Another approach is to use a kernel with exact and finite-dimensional feature map. Isolation Kernel Ting et al. (2018) is a such kernel which directly gets the feature map without calculating kernel matrix .

Let $D \subset \mathbb{R}^d$ be a dataset sampled from an unknown distribution $\mathbb{P}$. Isolation Kernel uses a partition mechanism such as Voronoi Diagram Qin et al. (2019); Zhang et al. (2023) or hypersphere Ting et al. (2020) to partition the data into $|\mathcal{D}|$ cells, where $|\mathcal{D}| = \psi$, and $\mathcal{D} \subset D$ is sampled from $D$ as the seeds of partition mechanism. Let $\mathbb{H}_\psi(D)$ denotes the set of all partitions $H$, each cell $\mathscr{I}[z]$ of partition $H$ isolates $z \in \mathcal{D}$ from the rest of the points in $\mathcal{D}$.

**Definition 4.3.** Isolation Kernel: $\forall x, y \in \mathbb{R}^d$, Isolation Kernel of $x$ and $y$ is defined to be the probability that $x$ and $y$ fall into the same cell $\mathscr{I}[z]$ of partition $H$ over all the partitions $H \in \mathbb{H}_\psi(D)$: $\kappa_I(x, y|D) = \mathbb{E}_{\mathbb{H}_\psi(D)}[\mathbf{1}(x, y \in \mathscr{I}[z]|\mathscr{I}[z] \in H)]$

Given the $t$ partitions $H_1, ..., H_t$, the feature map $\Phi(x)$ of $\kappa_I$ is a $\psi \times t$-dimensional binary column vector.

**Definition 4.4.** Isolation Kernel's feature map, denoted as $\Phi(x)$, is a vector that represents the cell in which $x$ falls into over partition of $t$ times. For $x \in \mathbb{R}^d$, the dimension of $\Phi(x)$ is $\psi \times t$, where $\psi$ is the number of cells in one partition $H$. Each element of $\Phi(x)$ is either 0 or 1, indicating the cell in which $x$ falls into.

Since the dimension of $\Phi(x)$ is finite, KME of $\kappa_I$ can be obtained by simply computing the mean of all feature maps.

Then $\mu_D$, $\mu_f$, and the parameters $\theta$ of model $A$ are concatenated as the input $\mathbf{x}$ of the neural network, which contains two hidden layers. The KME of distribution $\mathbb{P}$ on the given data $D$ is:

$$\mu_{\mathbb{P}} = \mathbb{E}[\kappa_I(\cdot, D)] = \frac{\sum_{x \in D} \Phi(x)}{|D|}, \tag{1}$$

where $\Phi(X)$ is IK's finite-dimensional feature map determined by two hyperparameters $t, \psi$. According to Equation 1, we have: $\mu_D = \frac{\sum_{x \in D} \Phi(x)}{|D|}$ and $\mu_f = \frac{\sum_{x \in D_f} \Phi(x)}{|D_f|}$. $\mu_D$ and $\mu_f$ are later concatenated with the parameters of the original model as input $\mathbf{x}$.

The output $\theta'$ is the parameter of the unlearned model $U(D, D_f, A(D))$. And Mean Squared Error (MSE) loss is used as the objective function for training $U$:

$$\mathcal{L} = \frac{1}{|\theta'|} \sum_{i=1}^{|\theta'|} (\theta'_i - \theta^r_i)^2, \tag{2}$$

where $\theta^r$ is the parameters of retrained model $A(D \setminus D_f)$.

---

[1]Other models can also be used. A two-layer (256, 64) fully connected neural network is used in our experiments.

How to Learn $U$

In order to train $U$, first we need a training set $\mathscr{D} = \mathscr{X} \times \mathscr{Y}$. We produce $\mathscr{D}$ by the following three steps. First, we generate $\mathfrak{D}$ by randomly sampling $s$ points from $D$ ($s \ll |D|$), and train a model $A(\mathfrak{D})$ on $\mathfrak{D}$. Second, we randomly select some samples as forget dataset $\mathfrak{D}_f \subset \mathfrak{D}$. $\mu_{\mathfrak{D}}, \mu_{\mathfrak{D}_f}$ of $\mathfrak{D}$ and $\mathfrak{D}_f$ are obtained using Equation 1. The parameter $\theta$ of model $A(\mathfrak{D})$ is concatenated with $\mu_{\mathfrak{D}}$ and $\mu_{\mathfrak{D}_f}$ as $\mathbf{x}$. Finally, we obtain $\theta^r$, the parameter of retrained model $A(\mathfrak{D} \setminus \mathfrak{D}_f)$. Now we have one instance $(\mathbf{x}, \theta^r) \in \mathscr{X} \times \mathscr{Y}$.

The above steps are repeated $m$ times to obtain $\mathscr{D}$ containing $m$ samples. $N_U$ is later trained on $\mathscr{D}$ by minimizing the loss function $\mathcal{L}$ shown in Equation 2. Since the parameters in KME do not require learning, learning $N_U$ is equivalent to learning $U$. When there is a $D_f$ that needs to be unlearned from model $A(D)$, we can obtain the unlearned model $A_{\theta'}$ via the learned $N_U$.

An important point to highlight is that once we have obtained $N_U$, if we need to unlearn any user data $D_f$, we can obtain the unlearned model $U(D, D_f, A(D))$ via the learned $N_u$ directly without having to repeat the preprocessing and training phase.

Generalization Bound Analysis

Here we provided the generalization bound of the classifier (Logistic Regression, Multilayer Perceptron) unlearned by L2UL.

Denote the expected loss of Logistic Regression model $A(x|\theta) = \frac{e^{\theta x + b}}{1 + e^{\theta x + b}}$ as

$$L(A) = E_{x,y}[y \log(A(x)) + (1 - y) \log(1 - A(x))].$$

L2UL learns the unlearning function $U$. The expected loss of the model unlearned by $U$ is upper-bounded, as shown in the following Theorem 4.5.

**Theorem 4.5.**
$$L(U(A(D), D, D_f)) \leq L(A(D \setminus D_f)) + 2R\sqrt{\epsilon},$$

*where $R$ is the radius of the dataset $X$ ($\forall x \in X, ||x|| \leq R$), and $\epsilon$ is the generalization error bound (MSE) of the unlearning model $U$ in L2UL.*

Similarly, the expected loss of Multilayer Perceptron (MLP) unlearned by $U$ is also upper-bounded, as shown in Theorem 4.6.

**Theorem 4.6.**
$$L(U(A(D), D, D_f)) \leq L(A(D \setminus D_f)) + C\sqrt{\epsilon},$$

*where $L$ is cross entropy loss, $C$ is a finite scalar determined by the MLP structure and $\epsilon$ is the generalization bound (MSE) of the unlearning model $U$ in L2UL.*

Both Theorem 4.5 and 4.6 indicate that the expected loss of the classifier unlearned by L2UL is close to that of retrained classifier.

# 5 Experiments

**System:** The experiments are executed on a Linux machine with 1T GB RAM and an AMD 128-core CPU, with each core running at 2 GHz.

**Data:** We use seven public datasets in our experimental evaluation [2]. The specifications of the datasets are summarised in Table 1.

**Comparison algorithms:** We compare L2UL [3] with Retrain, one $D$-oriented unlearning algorithm: SISA and two $A(D)$-oriented unlearning algorithms: DeltaGrad, FYEMU.

1. **Retrain:** As the most naive approach, it just retrains the model on the remaining dataset $D \setminus D_f$ to unlearn the information of $D_f$.

---

[2] The datasets are available at https://archive.ics.uci.edu/, and https://www.csie.ntu.edu.tw/~cjlin/libsvmtools/datasets/

[3] The codes are available at https://anonymous.4open.science/r/L2UL

Table 1: Dataset Summary. n=no.instances, d=no.attributes.

|   | Magic | Adult | Sepsus | Skin | Covetype | SUSY | HIGGS |
|---|---|---|---|---|---|---|---|
| n | 19,020 | 32,562 | 40,328 | 245,057 | 581,012 | 5,000,000 | 11,000,000 |
| d | 10 | 14 | 89 | 3 | 54 | 18 | 28 |

2. **SISA:** SISA Bourtoule et al. (2021) is a well-known framework that partitions the data to reduce the retraining time by just retraining a single model on the shard that needs to be forgotten.

3. **DeltaGrad:** DeltaGrad Wu et al. (2020) unlearns the data by differentiating the optimization path with the Quasi-Newton method based on information cached during the training phase.

4. **FYEMU:** FYEMU Tarun et al. (2023a) is a unlearning algorithm for Neural Networks, which first unlearns data by adding noise to the model parameters, and then obtains a new model through fine-tuning.

**Evaluation:** We evaluate the efficiency and effectiveness of unlearning algorithms using unlearning time and the accuracy of the unlearned classifier on test data (20%), which is averaged over 10 runs. In each run, we unlearn one randomly selected instance [4].

UNLEARN LINEAR CLASSIFIER: LOGISTIC REGRESSION

To begin, we assess the performance of our L2UL approach using a linear classifier known as Logistic Regression (LR). We have demonstrated that the generalization error of the unlearned classifier, denoted as $U(D, D_f, A(D))$, is bounded. Additionally, here we find that L2UL is both efficient and effective when applied to Logistic Regression.

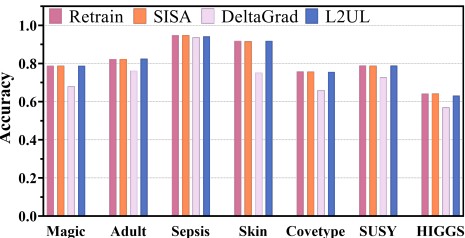 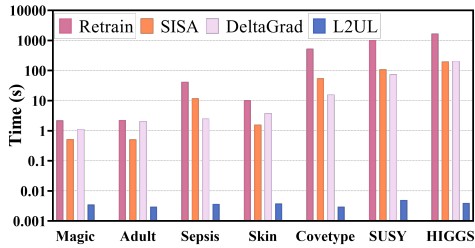

Figure 2: Results of unlearning Logistic Regression on seven datasets.

The results of unlearning one instance in terms of accuracy and unlearning time are shown in Figure 2. L2UL achieves comparable accuracy to Retrain and SISA and outperforms DeltaGrad over all seven datasets. Compared with Retain, SISA, and DeltaGrad, L2UL speeds up 630, 149, and 282 times respectively on the Magic dataset, and 422500, 49715, and 51500 times on the HIGGS dataset.

UNLEARN NON-LINEAR CLASSIFIER: MULTILAYER PERCEPTRON

Nonlinear classifiers can learn complex nonlinear boundary, which require a more complex process to learn. Besides, nonlinear classifier usually has more parameters, which makes their parameter space very large. Implementing unlearning in this large space is a more challenging problem.

In this subsection, we test the performance of L2UL on Multilayer Perceptron (MLP) Taud & Mas (2018). We use Cross Entropy as the loss of the MLP, which has 1 hidden layer with 10 neurons.

The results of unlearning one instance are shown in Figure 3. L2UL achieves comparable accuracy to Retrain and SISA except HIGGS (we will discuss this in Section D). L2UL achieves close accuracy to Deltagrad on Sepsis and Skin and outperforms DeltaGrad on five other datasets. L2UL outperforms

---

[4] $|D_f| = 1$ for LR, $|D_f| = 1$, $|D_f| = 100$, and $|D_f| = 1000$ for MLP.

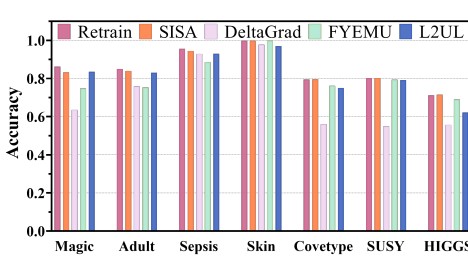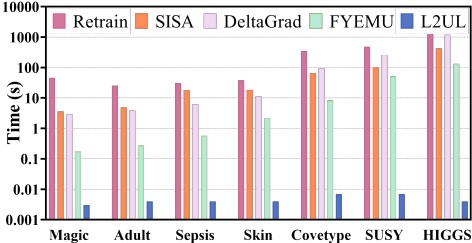

Figure 3: Results of unlearning MLP on seven datasets.

FYEMU on the two smallest datasets (Magic and Adult) and achieves close accuracy to FYEMU on five other datasets.

Compared with Retain, SISA, DeltaGrad, and FYEMU, L2UL speeds up 15087, 1203, 730, and 43 times respectively on the Magic dataset, and 398000, 105250, 294750, and 32750 times on the HIGGS dataset. The efficiency of L2UL comes from the simple structure of $U$, which ensures that a single run of $U$ is very quick. Moreover, once $U$ is learned, the parameters of $U$ do not need to change with high probability when new $D_f$ comes. We will discuss this in Section 6. L2UL just directly uses the learned $U$ to unlearn the new $D_f$, while the other algorithms require the costly computation again to unlearn $D_f$. This difference will be more significant when multiple unlearning requests come in sequence.

UNLEARN LARGE-SCALE PARAMETERS: RESNET

Furthermore, we verify the effectiveness of our proposed L2UL on a model with a larger parameter size. We tested the performance of unlearning ResNet-18 model on the CIFAR-10 dataset and compared it with:

**ADV+IMP:** Cha et al. (2024) Misclassify each instance outside of its original prediction, or relabel the instance to a different label. At the same time, use adversarial examples to overcome forgetting at the representation level and use weight importance indicators to accurately locate network parameters that propagate unnecessary information to reduce the time required for forgetting.

**RUM:** Zhao et al. (2024) The forgotten set is refined into homogenized subsets based on different features. The meta-algorithm employs the existing algorithms to forget each subset, ultimately providing a model that has forgotten the entire forgotten set.

We train ResNet-18 for 50 epochs on CIFAR-10 with a learning rate of 0.0001. And we report two evaluation metrics, ToW and MIA (Membership Inference Attack) gap, that are used in RUM Zhao et al. (2024).

Table 2: Results of unlearning ResNet-18 on CIFAR-10 dataset.

|  |  | 10 | 100 | 1000 | 3000 |
|---|---|---|---|---|---|
| ToW(↑) | Retrain | 1.00 | 1.00 | 1.00 | 1.00 |
|  | ADV+IMP | 0.32 | 0.13 | 0.01 | 0.01 |
|  | RUM | 0.60 | 0.8 | 0.83 | 0.83 |
|  | L2UL(Ours) | 0.80 | 1.00 | 0.88 | 0.99 |
| MIA gap(↓) | Retrain | 0.00 | 0.00 | 0.00 | 0.00 |
|  | ADV+IMP | 0.30 | 0.13 | 0.17 | 0.14 |
|  | RUM | 0.30 | 0.13 | 0.15 | 0.12 |
|  | L2UL(Ours) | 0.30 | 0.13 | 0.17 | 0.14 |
| Time(↓) | Origin ($A(D)$) | 1044.91 | - | - | - |
|  | Retrain | 1020.49 | 874.39 | 783.23 | 702.7 |
|  | ADV+IMP | 28.39 | 78.66 | 1665.48 | 12088.23 |
|  | RUM | 13.62 | 13.6 | 13.17 | 12.8 |
|  | L2UL(Ours) | 0.47 | 0.54 | 0.51 | 0.59 |

The results are shown in Table 2. We have the following three observations:

1. L2UL achieves the highest ToW scores when forgetting 10, 100, 1000, and 3000 samples.

2. L2UL and the comparison algorithms have a closed low MIA gap.

3. L2UL only takes a very short time to complete the unlearning.

UNLEARNING EFFICACY

In addition to effectiveness and efficiency, the machine unlearning algorithm must also be evaluated to determine whether it has forgotten users' information. We first show the two very commonly used evaluation metrics. Membership Inference Attack Chundawat et al. (2023a); Liu et al. (2024) (as used in Table 2) and accuracy on unlearned data. The results in terms of Membership Inference Attack and accuracy on $D_f$ ($|D_f|$=1000 for MLP) in Table 3 show that L2UL has forgotten the information of $D_f$ from $A(D)$[5]. The $U$ has learned the unlearning behaviors.

Table 3: Results in terms of Membership Inference Attack and accuracy on unlearn data.

| datasets | MIA | | | Accuracy on $D_f$ | | | |
|---|---|---|---|---|---|---|---|
| | Retrain | FYEMU | L2UL | Retrain | SISA | FYEMU | L2UL |
| Magic | 0.79 | 0.65 | 0.75 | 0.85 | 0.79 | 0.71 | 0.78 |
| Adult | 0.8 | 0.68 | 0.74 | 0.81 | 0.78 | 0.75 | 0.76 |
| Skin | 1 | 1 | 1 | 1 | 1 | 1 | 1 |
| Covetype | 0.73 | 0.68 | 0.67 | 0.82 | 0.76 | 0.76 | 0.71 |
| SUSY | 0.72 | 0.69 | 0.69 | 0.73 | 0.71 | 0.7 | 0.71 |
| HIGGS | 0.65 | 0.68 | 0.5 | 0.74 | 0.77 | 0.689 | 0.63 |

However, it is not the case that the lower these two scores are, the better the model performance is, because this may give rise to other issues, such as information exposure Golatkar et al. (2020) [6]. Therefore, we use an example to demonstrate the ability of the proposed algorithm L2UL for calibrating contaminated models, which also shows that L2UL has forgotten $D_f$.

As previously stated, training data can sometimes be contaminated, which negatively impacts the model's performance. Machine Unlearning is an effective method for cleaning the model when dirty data is detected. A simple example is shown in Figure 4(a). The artificial dataset has two classes, which can be classified by a linear model. But when the training set is contaminated (Figure 4(b) right shows an example with 50 contaminated points), the accuracy of the model will drop.

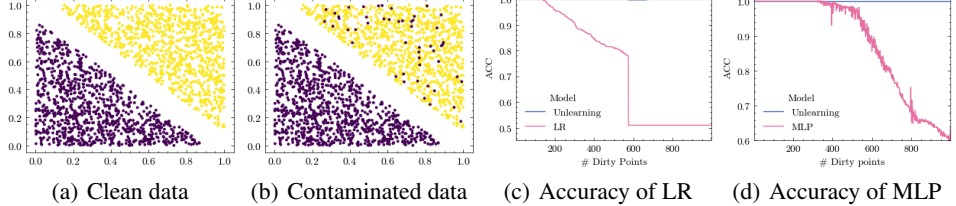

(a) Clean data     (b) Contaminated data     (c) Accuracy of LR     (d) Accuracy of MLP

Figure 4: Artificial dataset with clean data and contaminated data, and the results of LR and MLP.

Figure 4(c) shows the results of LR in terms of Accuracy. When data is not contaminated, the accuracy is 1, as expected. As the number of dirty points grows, the accuracy declines to approximately 0.5. When we use L2UL to clean the contaminated model by treating the dirty points as $D_f$, the accuracy of the unlearning model remains 1.00 as the number of dirty points grows. Similar results are shown in Figure 4(d) when we replace LR with MLP. The difference is that the accuracy and F1 of MLP decrease more slowly than LR. And accuracy and F1 of the unlearned model are always 1.00. This shows that L2UL has indeed achieved the unlearning of dirty data.

---

[5]L2UL classifies all samples into one class on Sepsis dataset, so MIA cannot be performed

[6]In this paper, if the scores are lower than both the original model ($A(D)$, ORI) and Retrain, it means the unlearning algorithm has forgotten the information.

# 6    DISSCUSSION

PARAMETER SENSITIVITY

L2UL is robust to the parameters $\psi$ and $m$ (The results on HIGGS are shown in Figure 9, more results are shown in the appendix). Compared to the parameters of the L2UL model and the parameters of IDK, L2UL requires more strong data as input. L2UL learns an unlearning function $U$, only if the input data $\mathscr{D}$ contains sufficient unlearning information, can a correct and good unlearning function be learned.

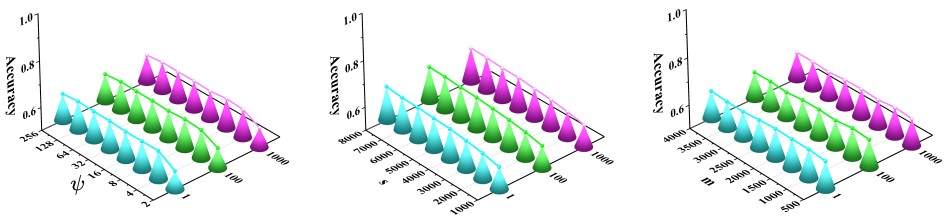

Figure 5: Parameter sensitivity analysis of $\psi$(left), $s$(middle), $m$(right) on HIGGS for MLP.

HOW TO LEARN $U$ WHEN THE MODEL CANNOT BE RETRAINED?

It works well for L2UL to use Equation 2 as its loss function only when the parameter of the retrained model $A(\mathfrak{D} \setminus \mathfrak{D}_f)$ can be obtained. However, the inability to retrain the model is a common challenge faced by machine unlearning. For example, we can't retrain model on streaming data.

In this case, L2UL can no longer be used to generate training data $\mathscr{D}$ and loss function 2 cannot be used to train $N_U$. How to find a new loss function when the model cannot be retrained is a challenge. we consider it as the future work of L2UL.

RETRAIN $U$ AFTER UNLEARNING $A$

It is worth emphasizing that when we employ $U$ to unlearn model, user's information $D_f$ will be included in $U$. Therefore, we must unlearn $U$ after we use $U$ to unlearn $A$.

**Theorem 6.1.** *When sampling $m$ subsets $\mathfrak{D}$ from $D$. The probability that $x$ is sampled at least $k_0$ times is*

$$\mathcal{P}(k \geq k_0) = \sum_{i=k_0}^{m} C_m^k (\frac{s}{|D|})^i (1 - \frac{s}{|D|})^{m-i}.$$

In our experiments, $\mathcal{P}(k \geq 2) < 0.004$ on HIGGS, which means the probability that the data $\{x, y\} \in D$ is used to train $U$ is very low. So we barely need to retrain $U$. Even if we need, the time cost is affordable, as it only takes 5.2 seconds to retrain $U$ on HIGGS.

# 7    CONCLUSION

To the best of our knowledge, L2UL is the first learning framework that learn the unlearning behaviors for machine unlearning, which is very simple but effective and efficient. We use logistic regression and multilayer perceptron as examples to experimentally demonstrate and theoretically analyze the effectiveness and efficiency of our method. L2UL is much faster than existing methods without sacrificing accuracy. An example of a contaminated model proves that our method can actually achieve unlearning. In addition, The results of the sensitivity analysis demonstrate that our approach remains stable when various hyperparameter configurations are used.

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

## A  LARGE LANGUAGE MODELS

We used Large Language Models to polish our writing.

## B  PSEUDO CODE OF L2UL

The pseudo code of L2UL is shown in Algorithm 1 and Algorithm 2.

---

**Algorithm 1:** Preprocessing

**Input**  : $D$: Dataset, $m$: Number of Samples in $\mathscr{D}$, $s$: size of $\mathfrak{D}$, $\psi$, $t$: Parameters of $\kappa_I$
**Output**: $\mathscr{D}$, $\Phi(\cdot)$

1  $\mathscr{D} = \{\}$;
2  Get the map function $\Phi(\cdot)$ using $\kappa_I$. ;
3  $\forall x \in D$, get the feature map $\Phi(x)$;
4  **for** $i \leftarrow 1$ **to** $m$ **do**
5  $\quad$ $\mathfrak{D} \leftarrow$ randomly sample $s$ points from $D$;
6  $\quad$ $\mathfrak{D}_f \leftarrow$ randomly sample from $\mathfrak{D}$ ;
7  $\quad$ $A(\mathfrak{D}), A(\mathfrak{D} \setminus \mathfrak{D}_f) \leftarrow$ train A on $\mathfrak{D}, \mathfrak{D} \setminus \mathfrak{D}_f$;
8  $\quad$ $\theta, \theta^r \leftarrow$ Extract parameters of $A(\mathfrak{D}), A(\mathfrak{D} \setminus \mathfrak{D}_f)$;
9  $\quad$ $\mu_\mathfrak{D}, \mu_f \leftarrow$ obtain KME using Equation 1 ;
10 $\quad$ $\mathbf{x} \leftarrow$ concatenation of $\mu_\mathfrak{D}, \mu_f$ and $\theta$ ;
11 $\quad$ $\mathscr{D} \leftarrow \mathscr{D} \cup \{(\mathbf{x}, \theta^r)\}$;
12 **end**
13 Return $\mathscr{D}$, $\Phi(\cdot)$.

---

**Algorithm 2:** Unlearn $D_f$

**Input**  : $N_U, \mu_D, \Phi(\cdot), A(D)$
**Output**: $A_{\theta'}$

1  $\forall x \in D_f$, get the feature map $\Phi(x)$;
2  $\mu_f \leftarrow$ obtain KME of $D_f$ through Equation 1;
3  $\mathbf{x} \leftarrow$ concatenation of $\mu_D, \mu_f$ and $A(D)$'s parameter $\theta$ ;
4  $\theta' \leftarrow N_U(\mathbf{x})$;
5  Return unlearned model $A_{\theta'}$.

---

## C  PROOFS

PROOF OF THEOREM 4.5

**Theorem 4.5**

$$L(f_{U(A(D),D,D_f)}) \leq L(f_{A(D \setminus D_f)}) + 2R\sqrt{\epsilon},$$

where $R$ is the radius of the dataset $X$ ($\forall x \in X, ||x|| \leq R$), and $\epsilon$ is the generalization bound (MSE) of the unlearning model $U$.

*Proof.* Denote $U(A(D), D, D_f) - A(D \setminus D_f)$ as $e$, then $||e|| \leq \sqrt{\epsilon}$. By Taylor Expansion, we approximately have

$$L(f_{U(A(D),D,D_f)}) - L(f_{A(D \setminus D_f)})$$

$$= \frac{\partial L(f_{A(D \setminus D_f)})}{\partial A(D \setminus D_f)}^\top e + o(||e||),$$

where $\lim_{\|e\|\to 0}\frac{o(\|e\|)}{\|e\|}=0$. Then by Cauchy–Schwarz inequality and convexity of norm function, we have

$$L(f_{U(A(D),D,D_f)})-L(f_{A(D\setminus D_f)})$$
$$=E_{x,y}[(f_{A(D\setminus D_f)}(x)-y)x^\top]e+o(\|e\|)$$
$$\leq\|E_{x,y}[(f_{A(D\setminus D_f)}(x)-y)x]\|\cdot\|e\|+o(\|e\|)$$
$$\leq E_{x,y}[\|(f_{A(D\setminus D_f)}(x)-y)x\|]\cdot\|e\|+o(\|e\|)$$
$$\leq 2R\sqrt{\epsilon}+o(\sqrt{\epsilon}).$$

$\square$

Here we refer $U(A(D),D,D_f)$ and $A(D\setminus D_f)$ as the parameters of the unlearned model and retrained model, respectively.

PROOF OF THEOREM 4.6

**Theorem 4.6**
$$L(f_{U(A(D),D,D_f)})\leq L(f_{A(D\setminus D_f)})+A\sqrt{\epsilon},$$

where $A$ is a finite scalar determined by the MLP structure and $\epsilon$ is the generalization bound (MSE) of the unlearning model $U$.

*Proof.* For the $(N+1)$-layer MLP model (the 0-th layer is the input layer, the $(N+1)$-th layer is the output layer), we use softmax activation for the output layer, and sigmoid activation function for the rest layers. We denote the number of neurons in the i-th layer as $n_i$, then the weight between $(i-1)$-th layer and $i$-th layer are $W_i$ of shape $(n_k,n_{k-1})$ and $B_i$ of shape $(n_i,1)$. For a finite test set $X,|X|=m$, we denote it as $A_0$. Then for the forward propagation scheme is described as following:

$$Z_k=W_kA_k+B_k,\ A_k=f_k(Z_k)$$

for $k\in\{1,2,...,N+1\}$, where $f_k(\cdot)$ is the activation function we use in k-th layer (softmax for the output layer, sigmoid for the rest layers).

For the back propagation, when using Cross Entropy Loss, we have

$$d\,Z_{N+1}=A_{N+1}-Y,\ d\,W_{N+1}=\frac{1}{m}d\,Z_{N+1}A_N^\top,$$

$$d\,B_{N+1}=\frac{1}{m}d\,Z_{N+1}1_{(m,1)},$$

and

$$d\,Z_k=W_{k+1}^Td\,Z_{k+1}\odot f_k'(Z_k),\tag{3}$$

$$d\,W_k=\frac{1}{m}d\,Z_kA_{k-1}^\top,\ d\,B_k=\frac{1}{m}d\,Z_k1_{(m,1)}$$

for $k\in\{1,2,...,N,N+1\}$, where $d$ means the derivative of the Cross Entropy Loss, $1_{(m,1)}$ is a column vector of size m, with each entry $=1$, and $\odot$ is the element-wise multiplication.

Denote $A(D\setminus D_f)$ as $\Theta$ and $U(A(D),D,D_f)$ as $\hat{\Theta}$. We have

$$\Theta=\begin{bmatrix}W\\B\end{bmatrix},\tag{4}$$

where $W,B$ are the concatenation of each layer's row-flatten parameter, i.e. $W=[\text{rft}(W_1),...,\text{rft}(W_{N+1})]^\top,W=[\text{rft}(B_1),...,\text{rft}(B_{N+1})]^\top$, where $\text{rtf}(\cdot)$ is the operator that flatten a matrix into a row vector. The same notation goes for $\hat{\Theta}$. As in 4.5, we only need to show that $d\,\Theta$ is bounded, which can be achieved from the bound for $d\,B$ and $d\,W$, since $\|d\,\Theta\|^2=\|d\,W\|^2+\|d\,B\|^2$.

For $W_{N+1}$, we have

$$\|d\,\text{rft}(W_{N+1})\|^2=\|\frac{(A_{N+1}-Y)A_N^\top}{m}\|_F^2\leq 4n_Nn_{N+1}.$$

since the absolute value of each entry in $A_{N+1}, A_N, Y$ is smaller than 1. For the bound on $d\,\mathrm{rft}W_k(k \neq N+1)$, we first need to get the bound $R_k$ for $d\,\mathrm{rft}Z_k$. We know that

$$\|d\,\mathrm{rft}(Z_{N+1})\|_\infty \leq R_{N+1} = 2.$$

Then according to Equation 3, we have

$$\|d\,\mathrm{rft}Z_k\|_\infty \leq \frac{R}{4}n_{k+1}R_{k+1} \triangleq R_k,$$

under the assumption $\|W\|_\infty \leq R$. The iterative form of $R_k$ can be expressed equivalently as

$$R_k = (\frac{R}{4})^{N+1-k}\prod_{j=k+1}^{N+1} n_j.$$

Coming back to $W_k$, for $k \leq N$, we have

$$\|d\,\mathrm{rft}(W_k)\|^2 \leq R_k^2 n_k n_{k-1}.$$

Finally, we have

$$\|d\,\mathrm{rft}W\|^2 = \sum_{k=1}^{N+1}\|d\,\mathrm{rft}W_k\|^2$$

$$\leq 4n_N n_{N+1} + \underbrace{\sum_{k=1}^{N}(\frac{R^2}{16})^{N+1-k}\frac{\prod_{j=k-1}^{N+1} n_j^2}{n_k n_{k+1}}}_{\triangleq A_W}.$$

Similarly, we have

$$\|d\,\mathrm{rft}(W_k)\|^2 \leq R_k^2 n_k n_{k-1}. \tag{5}$$

Finally, we have

$$\|d\,\mathrm{rft}B\|^2 = \sum_{k=1}^{N+1}\|d\,\mathrm{rft}B_k\|^2$$

$$\leq 4n_{N+1} + \underbrace{\sum_{k=1}^{N}(\frac{R^2}{16})^{N+1-k}\frac{\prod_{j=k}^{N+1} n_j^2}{n_k}}_{\triangleq A_B}. \tag{6}$$

Then $\|d\,\Theta\| \leq \sqrt{A_w^2 + A_B^2} \triangleq A$. $\qquad\square$

PROOF OF COMPLEXITY

* Time complexity of preprocessing is $\mathcal{O}(\psi t|D|d + m\mathcal{T}(A(s)))$

  *Proof.* Preprocessing first requires a total time of $\mathcal{O}(\psi t|D|d$ to produce the feature map (line 2 in Algorithm 1), and requires $\mathcal{O}(\psi ts + \mathcal{T}(A(s))$ for each loop, So the time complexity of Algorithm 1 is $\mathcal{O}(\psi t|D|d + m\mathcal{T}(A(s)))$. $\qquad\square$

* Time complexity of training $U$ is $\mathcal{T}(U(m))$.

  *Proof.* The time complexity of Train U is using the L2UL to train $N_U$ on the $m$ data points, so the time complexity is $\mathcal{T}(U(m))$. $\qquad\square$

* Time complexity of unlearning $D_f$ from $U$ is $\mathcal{O}(\psi t|D_f|d)$.

  *Proof.* Algorithm 2 requires time of $\mathcal{O}(\psi t|D_f|d)$ to get the feature map (line 1). and $\mathcal{O}(\psi t|D_f|)$ to get $\mu_f$ (line 2). Therefore, the time complexity to unlearn $D_f$ from the $U$ is $\mathcal{O}(\psi t|D_f|d)$. $\qquad\square$

* Space complexity of $U$ is $\mathcal{O}(\psi td)$.

  *Proof.* we need $\mathcal{O}(\psi t)$ to store the $\mu_D$, and $\mathcal{O}(\psi td)$ to store the seeds for partition in isolation kernel. Threefore, the space complexity of $U$ is $\mathcal{O}(\psi td)$. $\qquad\square$

PROOF OF THEOREM 7.1

**Theorem 7.1**

We sample $m$ subsets $\mathfrak{D}$ from $D$. The probability that sample $x$ is sampled at least $k_0$ times is:

$$\mathcal{P}(k \geq k_0) = \sum_{i=k_0}^{m} C_m^k (\frac{s}{|D|})^i (1 - \frac{s}{|D|})^{m-i}.$$

*Proof.* Sampling $s$ points from $|D|$ points, The probability that point $x$ is sampled is $\mathcal{P}(x) = \frac{s}{|D|}$. Do this sampling $m$ times, the probability that point $x$ is sampled $k$ times is:

$$P(k) = C_m^k (\frac{s}{|D|})^k (1 - \frac{s}{|D|})^{m-k}.$$

So the probability that sample $x$ is sampled at least $k_0$ times is:

$$\mathcal{P}(k \geq k_0) = \sum_{i=k_0}^{m} C_m^k (\frac{s}{|D|})^i (1 - \frac{s}{|D|})^{m-i}.$$

$\square$

## D  ADDITIONAL EXPERIMENTS

RESULTS OF UNLEARNING 100 AND 1000 INSTANCES IN MLP.

The results of unlearning 100 instances for MLP are shown in Figure 6.

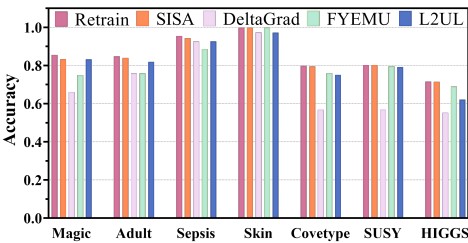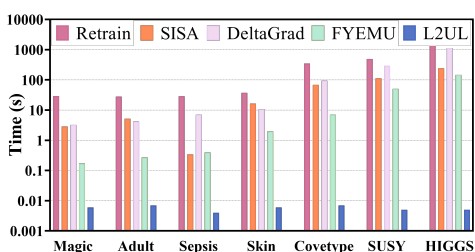

Figure 6: Results of unlearning MLP on seven datasets (# unlearning instance=100).

The results of unlearning 1000 instances for MLP are shown in Figure 7.

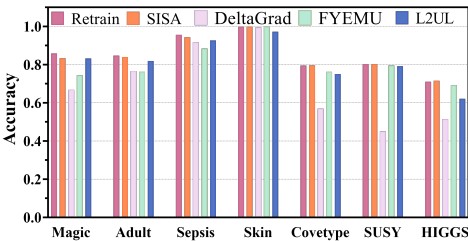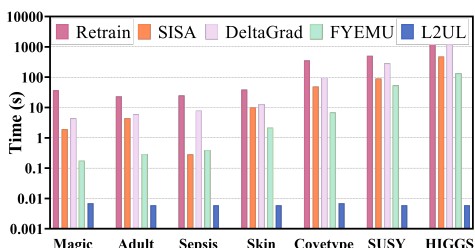

Figure 7: Results of unlearning MLP on seven datasets (# unlearning instance=1,000).

PARAMETER SENSITIVITY ANALYSIS

There are four hyperparameters in the Preprocessing and Unlearning phase of L2UL: $\psi, t$ for Isolation Kernel and $s, m$ for generating $\mathscr{X} \times \mathscr{Y}$. In our experiment, the default setting of hyperparameters is: $\psi = 4$, $t = 100$, $m = 1000$, $s = 1000$ for LR, and $s = 3000$ for MLP.

We report the accuracy of unlearning 1, 100, and 1,000 instances from LR and MLP on the largest datasets HIGGS, with $\psi \in [2, 4, 8, 16, 32, 64, 128, 256]$Ting et al. (2020). Ting et al. (2020) shows that the Isolation Distritbuion Kernel is not sensitive to $t$, hence, the sensitivity analysis of $t$ is omitted here.

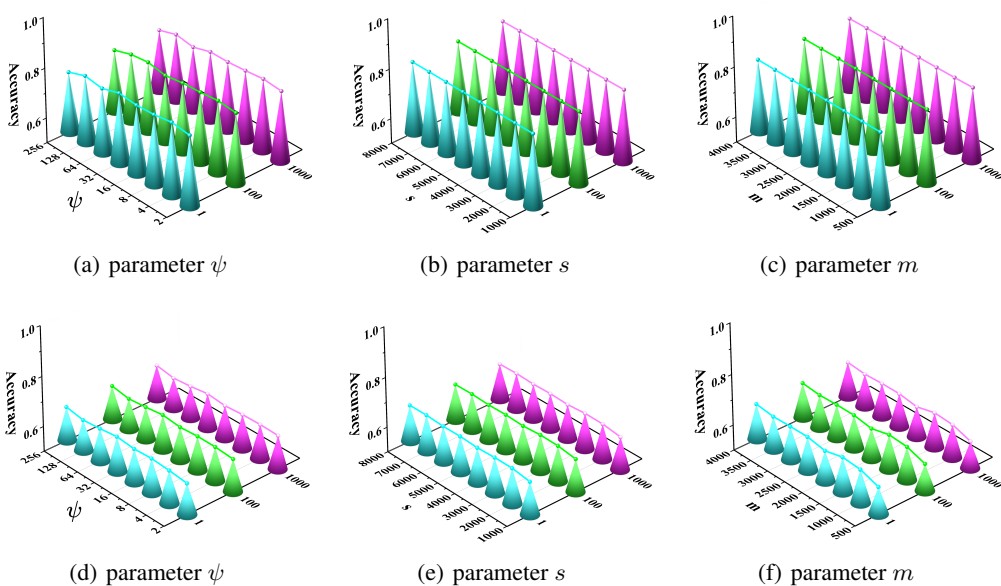

Figure 8: Parameter sensitivity analysis of LR on SUSY (top) and HIGGS (bottom).

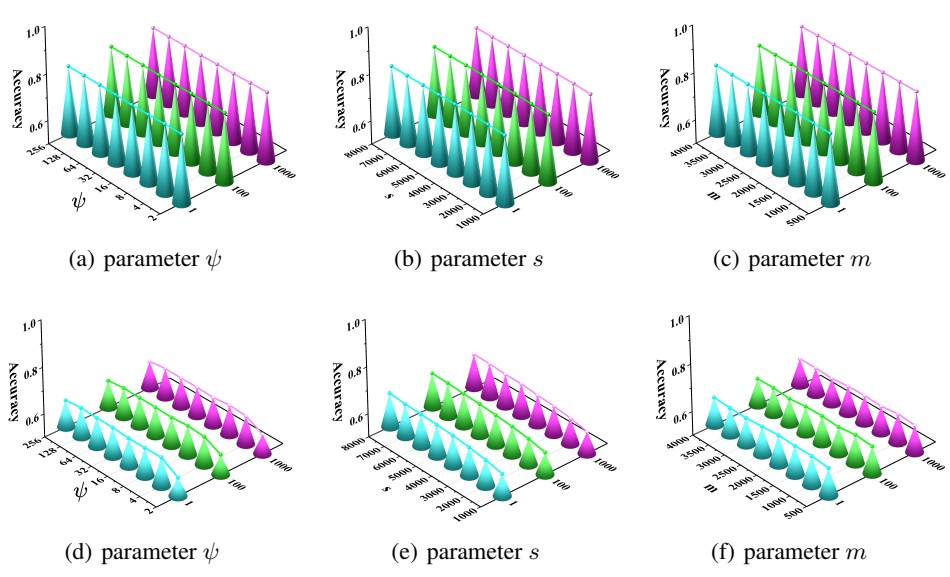

Figure 9: Parameter sensitivity analysis of $\psi$(left), $s$(middle), $m$(right) on SUSY (top) and HIGGS (bottum) for MLP.

Table 4: Factor of time complexity ($D$-or: $D$-oriented, $D_f$-or: $D_f$-oriented, $A(D)$-or: $A(D)$-oriented).

| $U$ | $D$-or | $D_f$-or | $A(D)$-or | L2UL |
|---|---|---|---|---|
| Factor | $A\&D$ | $D_f$ | $D_f\&\theta$ | $D_f$ |

The results of parameter sensitivity are shown in Figure 8 and Figure 9. L2UL is robust to $\psi$ when $\psi \geq 4$. The results of $s$ are shown in Figure 9(e). For LR on SUSY, HIGGS, and MLP on SUSY, L2UL is robust to $s$, and the accuracy of L2UL is close to that of Retrain. For MLP on HIGGS, as $s$ increases, the accuracy of L2UL increases from 0.60 to 0.65. This is because when $s$ (sample size) is small, we cannot get a good model $A(\mathfrak{D})$ and $A(\mathfrak{D} \setminus \mathfrak{D}_f)$ as input into L2UL. When sample size $s$ increases, we can have a better model $A(\mathfrak{D})$ and $A(\mathfrak{D} \setminus \mathfrak{D}_f)$, so that a good $U$ and unlearned model $U(D, D_f, A(D))$ with higher accuracy can be obtained on HIGGS.

L2UL is robust to $m$, as $N_U$ is a simple model that can achieve good performance with a small dataset $\mathscr{D}$.

In a nutshell, L2UL is robust to the parameters $\psi$ and $m$. Compared to the parameters of the L2UL model and the parameters of IDK, L2UL requires more strong data as input. L2UL learns an unlearning function $U$, only if the input data $\mathscr{D}$ contains sufficient unlearning information [7], can a correct and good unlearning function be learned.

# E    COMPLEXITY ANALYSIS

We show the factor that affects time complexity of unlearning algorithms as follows (time complexity of the machine learning model $A$ is denoted as $\mathcal{T}(A)$) :

1. $D$-oriented unlearning algorithm, such as SISA Bourtoule et al. (2021), has a time complexity of $\mathcal{T}(A(|D|/k))$, where $k$ is the number of shards. Because SISA retrains $A$ on some shards, the time complexity is limited by $A$ and $|D|$.

2. $D_f$-oriented unlearning algorithm, such as Guo et al. (2020), considers the influence of $D_f$, whose time complexity is $\mathcal{O}(|D_f|d^3)$, which is limited by $D_f$.

3. $A(D)$-oriented unlearning algorithm, such as Sekhari et al. (2021), considers how to update parameters. Its time complexity is $\mathcal{O}(|D_f|d^2 + d^{|\theta|})$ and is limited by $D_f$ and $|\theta|$.

The factors of time complexity are summarized in Table 4. Intuitively, $D_f$ is unavoidable because we at least need to know the samples we need to forget. L2UL is also subject to this restriction. Given dataset $D$, we give the time complexity and space complexity of L2UL as follows:

* Time complexity of preprocessing is $\mathcal{O}(\psi t|D|d + m\mathcal{T}(A(s)))$
* Time complexity of training $U$ is $\mathcal{T}(U(m))$.
* Time complexity of unlearning $D_f$ from $U$ is $\mathcal{O}(\psi t|D_f|d)$.
* Space complexity of $U$ is $\mathcal{O}(\psi td)$.

When unlearning $D_f$, L2UL needs to calculate the feature map of $D_f$ first. But we can store the feature map of every point from $D$ in advance. In this way, when unlearning $D_f$, we only need to select the corresponding feature maps and take the mean. Although the time complexity still depends on $D_f$, the operation of just taking mean is very fast. As a trade-off, space complexity goes up to $\mathcal{O}(\psi t|D|)$.

# F    KERNEL MEAN EMBEDDING

Kernel mean embedding is a nonparametric method which uses an element of a Reproducing Kernel Hilbert Space (RKHS) as the representation of a probability distribution. The mean feature map

---

[7]Sufficient unlearning information means a correct and well-trained model $A(\mathfrak{D})$ and a correct and well-unlearned model $A(\mathfrak{D} \setminus \mathfrak{D}_f)$.

$\mu_{\mathbb{P}} = \mathbb{E}[\kappa(\cdot, D)]$ of points $X$ based on kernel $\kappa$ embeds distribution $\mathbb{P}$ into RKHS, and preserve all of the statistical features of $\mathbb{P}$.

In our framework, when using some common kernel such as Gaussian kernel, Laplacian kernel, etc. $\mu_{\mathbb{P}}$ will map $\mathbb{P}$ into a infinite-dimensional space, which cannot be used as input to the neural network. In other words, we need a kernel whose RKHS dimension is finite.

One approach is to use low-rank representations of kernel matrix. The most popular examples are Nyström method Kumar et al. (2012); Williams & Seeger (2000); Musco & Musco (2017) and the Random Fourier Features Rahimi & Recht (2007); Yang et al. (2012). However, these are all approximate methods and will reduce effectiveness.

Another approach is to use a kernel with exact and finite-dimensional feature map. Isolation Kernel Ting et al. (2018) is a such kernel which directly gets the feature map without calculating kernel matrix .

Let $D \subset \mathbb{R}^d$ be a dataset sampled from an unknown distribution $\mathbb{P}$. Isolation Kernel uses a partition mechanism such as Voronoi Diagram Qin et al. (2019); Zhang et al. (2023) or hypersphere Ting et al. (2020) to partition the data into $|\mathcal{D}|$ cells, where $|\mathcal{D}| = \psi$, and $\mathcal{D} \subset D$ is sampled from $D$ as the seeds of partition mechanism. Let $\mathbb{H}_\psi(D)$ denotes the set of all partitions $H$, each cell $\mathscr{I}[z]$ of partition $H$ isolates $z \in \mathcal{D}$ from the rest of the points in $\mathcal{D}$.

**Definition F.1.** Isolation Kernel: $\forall x, y \in \mathbb{R}^d$, Isolation Kernel of $x$ and $y$ is defined to be the probability that $x$ and $y$ fall into the same cell $\mathscr{I}[z]$ of partition $H$ over all the partitions $H \in \mathbb{H}_\psi(D)$:
$$\kappa_I(x, y|D) = \mathbb{E}_{\mathbb{H}_\psi(D)}[\mathbb{1}(x, y \in \mathscr{I}[z]|\mathscr{I}[z] \in H)]$$

Given the $t$ partitions $H_1, ..., H_t$, the feature map $\Phi(x)$ of $\kappa_I$ is a $\psi \times t$-dimensional binary column vector.

**Definition F.2.** Isolation Kernel's feature map, denoted as $\Phi(x)$, is a vector that represents the cell in which $x$ falls into over partition of $t$ times. For $x \in \mathbb{R}^d$, the dimension of $\Phi(x)$ is $\psi \times t$, where $\psi$ is the number of cells in one partition $H$. Each element of $\Phi(x)$ is either 0 or 1, indicating the cell in which $x$ falls into.

Since the dimension of $\Phi(x)$ is finite, KME of $\kappa_I$ can be obtained by simply computing the mean of all feature maps.

