# OpenReview forum: "Learning to Unlearn: Machine Unlearning via Learning the Unlearning Behaviors"
_ICLR.cc/2026/Conference — Submitted to ICLR 2026_

### Official Review · Reviewer_gR2i · 2025-10-17

**Soundness:** 3
**Presentation:** 2
**Contribution:** 2
**Rating:** 2
**Confidence:** 4

**Summary:**

Summary:
This paper proposes a learning based model-agnostic approach (L2UL) based on a distribution perspective to implement unlearning. Some comments are provided as follows.

**Strengths:**

Strengths:
1. The paper proposed to implement unlearning through learning, which is interesting.

2. The problem is clearly defined and provided some analysis.

3. Extensive experiments are conducted in different datasets.

**Weaknesses:**

Weaknesses:
1. In Eq.(2), we need to train U with the parameters of retrained model $\theta^r$. However, if we have $\theta^r$, we already have the unlearned model. Why do we need to do other training to implement unlearning.

2. The advantages are overcliamed, it says "L2UL achieves unlearning through learning rather than designing." However, to achieve unlearning, L2UL also designs many components.

3. The advantage says "the execution time of unlearning via the learned U is very short." However, to achieve unlearning, L2UL needs to sample many x and to retrain many corresponding $\theta^r$, which is very computationally expensive.

4. The writing needs to improve, too many short (two lines) paragraphs.

5. L2UL makes unlearning more complex as it need to train U to unlearn then must unlearn U after unlearning.

**Questions:**

See weaknesses.

**Details Of Ethics Concerns:**

No ethics concerns

---

> ### Author Response · Authors · 2025-11-20
> **rebuttal**
>
> Thank you for your review and suggestions.
>
> A1. Because retraining on the entire dataset is costly, our method only retrains on a small sampled portion of the dataset.
>
> A2. We learn the unlearning behavior for unlearning, but that doesn't mean we don't use any components at all.
>
> A3. We only need to train $U$ once, and use this trained $U$ every time we forget. We don't need to retrain $U$ for subsequent forgetting events.
>
> A4. Thank you very much. We have merged these short paragraphs.
>
> A5. As we stated in the paper, forgetting $U$ is not necessary every time we need to forget. When we do need to unlearn $U$, forgetting $U$ is also very fast, for example, it only takes 5.2 seconds on HIGGS.

---

### Official Review · Reviewer_vrxN · 2025-10-30

**Soundness:** 2
**Presentation:** 3
**Contribution:** 2
**Rating:** 2
**Confidence:** 4

**Summary:**

The paper introduces Learning-to-UnLearn (L2UL), a learning-based and model-agnostic approach to machine unlearning. It aims to match the retrained model while avoiding expensive retraining. The method learns an unlearning function that maps three inputs to updated parameters: a representation of the full training set, a representation of the forget set, and the original model parameters. The authors provide bounds showing that the expected loss of the unlearned model is close to that of the retrained model for logistic regression and MLP. Experiments show accuracy comparable to retraining and strong efficiency on tabular datasets. The approach is also tested on ResNet-18 for CIFAR-10 to illustrate scalability.

**Strengths:**

S1. The paper provides a theoretical generalization bound for the proposed Learning-to-Unlearn (L2UL) model from the loss perspective, offering a sound theoretical foundation.

S2. On logistic regression and MLP tasks, L2UL achieves test accuracy close to retraining while requiring far less time than existing methods such as SISA, DeltaGrad, and F–YEMU.

S3. The presentation is clear. Figures and definitions effectively illustrate the method and make its motivation and design easy to understand.

**Weaknesses:**

W1. Although a generalization bound is provided, it is derived only from the loss level and applies mainly to simple models such as logistic regression and MLP. Compared with certified unlearning methods that give guarantees at the parameter level, this bound is relatively weak and does not ensure full effectiveness of unlearning.

W2. The experiments rely mainly on relatively simple tasks, raising concerns about scalability. Even though ResNet results are included, the paper does not compare against strong approximate unlearning baselines such as SalUn[Fan+2024], Boundary Unlearning[Chen+2024], and SFRON[Huang+2024], so its unique advantage remains unclear.

**References**
[Fan+2024] Fan, C., et al. SalUn: Empowering Machine Unlearning via Gradient-based Weight Saliency in Both Image Classification and Generation. ICLR 2024.
[Chen+2024] Chen, M., et al. Boundary Unlearning: Rapid Forgetting of Deep Networks via Shifting the Decision Boundary. CVPR 2024.
[Huang+2024]Huang, Z., et al. Unified Gradient-based Machine Unlearning with Remain Geometry Enhancement. NeurIPS 2024.

W3. The ResNet experiments also have several issues:

- (1) The reported runtime seems to reflect only inference time, not the full cost of constructing and training the L2UL dataset, which requires repeated retraining.

- (2) Important experimental details are missing, such as the number of samples used for the L2UL training dataset on CIFAR-10, the number of datasets constructed, and the optimizer used with the learning rate of 0.0001.

W4. The parameter sensitivity analysis is limited to a single setup (HIDDS and MLP), which reduces its practical value. Evaluation in logistic regression and MLP experiments relies only on test accuracy, which is insufficient to fully measure unlearning effectiveness.

W5. Presentation issues include missing citations for GDPR, CCPA, PIPEDA, and some prior unlearning methods, as well as very short paragraphs (1–2 sentences) in the Introduction that affect readability.

**Questions:**

Q1. In Theorems 4.5 and 4.6, could you clarify why having a bound on the loss implies that unlearning is successful? It is not fully clear to me how this connection is established. Also, are the constants R and C in these theorems themselves bounded? What is the approximate scale of \epsilon in your bound?

Q2. Did you report the time required to construct the training dataset for L2UL? This step seems computationally expensive, but the paper does not specify it.

Q3. In the ResNet experiment, you used a learning rate of 0.0001. Could you specify which optimizer was used? The sentence describing this part is unclear, and the setting “apples = 50” also seems unreasonable. Could you clarify what this means?

Q4. For the CIFAR-10 experiment, please provide more details on how the L2UL training dataset was constructed. What parameters were used? How large was each sub-dataset, and how many training–retraining pairs were used to train L2UL? These details appear to be missing.

Q5. In the parameter sensitivity studies, it would be more convincing to include experiments on more realistic models such as ResNet, rather than only MLP.

Q6. On line 935, you mention “the parameters of IDK.” What does IDK stand for? It appears twice in the paper but is never defined.

---

> ### Author Response · Authors · 2025-11-20
> **rebuttal**
>
> Thank you for your review and suggestions.
>
> A1. We want the performance of the forgotten model to be decent, ideally close to that of the retrained model. $R$ and $C$ are both bounded constants, and $\epsilon$ represents a very small error.
>
> A2. Since we only retrain on a small number of sampled points, this part of the time is not particularly expensive, and we only need to prepare once, not every time we forget the data.
>
> A3. We used the Adam optimizer and fine-tuned the epoch to 50 after importing the pre-trained model.
>
> A4. We set $\psi=4$, $t=100$, $m=1000$, $s=3000$.
>
> A5. Our results on LR and MLP are consistent, so we did not repeat the experiments on ResNet.
>
> A6. IDK refers to the isolation distribution kernel we use. Thank you very much for pointing this out; we have now added this in page 4.

---

### Official Review · Reviewer_NnBA · 2025-10-31

**Soundness:** 3
**Presentation:** 3
**Contribution:** 3
**Rating:** 4
**Confidence:** 3

**Summary:**

This paper proposes a new framework (Learn-to-UnLearn) that allows automatically learning the unlearning strategy using the optimization methods. Learn-to-UnLearn uses neural network architecture to output the unlearned model from the input encompassing 3 components: the model A(D) learned from full training data, the representations for the distribution of forget dataset D_f and the original dataset D. The authors already provided theoretical results to support the effectiveness of their unlearning method under setting of Logistic Regression and Multi-Layer Perceptron. Empirical evaluation was also conducted in diverse setting to show competitive results of the proposed framework.

**Strengths:**

1. The paper is well-written with clear motivation.
2. The authors propose an innovative idea to build unlearning strategy as a neural network, which can be trained once we already have retrained models as training data. This combination may open a new direction in finding unlearning algorithm U effectively.
3. Theorems and proofs were provided to support theoretical aspects of the proposed framework under simple settings.
4. Diverse experiments including diverse comparisons would greatly support empirical aspects of the proposed approach. The authors also provide a setting to unlearn large-scale parameters, which promises well for Learn-to-UnLearn's performance in practical contexts.

**Weaknesses:**

1. Lack of details for network architecture of unlearning method U, as well as the comparison between different architectures when performing unlearning tasks.

2. Different datasets will result in distinct learned model A(D)'s behaviors and dataset's distribution, these factors will significantly affect the input and output domain of the unlearning method U. Therefore, I raise the following concerns:
i) In the section 4, 'HOW TO LEARN U', the authors only train U with small subsets of D (size s << |D|). Since these subsets are significantly small, they cannot assure the same behaviors as the full dataset D. Therefore, it limits scalability and generalization with huge datasets when there is a discrepancy between distribution of set used in training U and test set in practice.
ii) Method U should be trained separately with different datasets (e.g, CIFAR-10, HIGGS,...), it means that training time of U would be included in comparisons with other baselines (Other baselines do not require to train U for each dataset, so they do not need to include this information). However, the authors did not show the training time for every dataset in the experiments, they only included the inference time of U.

3. In figure 3, when compared with other baselines on HIGGS, performance of L2UL is significantly (10%) lower than other baselines (SISA, FYEMU). These results show the huge degradation in performance of L2UL on larger datasets and Model A(D). From this insight, I am concerned about the scalability of the proposed method as the training cost and scale of U also increase significantly.

**Questions:**

From above weakness, I recommend the authors to:
1. Provide network architecture used for U and comparisons between different architectures.
2. Clarify the training time of U in every setting and use it along with inference time to compare the performance with other baselines.
3. Explain the scalability of the proposed method (Is it still effective and practical when the model A(D) and datasets become significantly larger?) and the discrepancy mentioned in the weaknesses.

---

> ### Author Response · Authors · 2025-11-20
> **rebuttal**
>
> Thank you for your review and suggestions.
>
> A1. In our experiments, we used a two-layer (256,64) fully connected neural network. We found that this simple model can effectively achieve forgetting. We will add analyses of different architectures in the future, such as networks of various depths and dimensions.
>
> A2. Thank you for your suggestion. We mainly compared the forgetting time for each algorithm because, in machine learning, people are more concerned about the time required for forgetting. We trained $U$ on HIGGS in only 5.2 seconds, which is even less than the forgetting time of many algorithms. This is because the architectures we used are very simple.
>
> A3. Thank you very much for your suggestion. We have already validated our algorithm on relatively large datasets, such as HIGGS. For larger models, we have only conducted experiments on ResNet-18 so far. However, due to our equipment limitations, we have not performed validation on even larger models, which is a shortcoming of ours. We will emphasize this in our paper. For models that are not particularly large, we recommend that users use our algorithm. For larger models that we have not yet validated, we recommend that users use the algorithms that have already been validated.

---

### Official Review · Reviewer_pEsS · 2025-11-01

**Soundness:** 2
**Presentation:** 3
**Contribution:** 2
**Rating:** 4
**Confidence:** 4

**Summary:**

This paper proposes L2UL (Learning-to-UnLearn), a learning-based approach to machine unlearning that uses a neural network to learn an unlearning function U. The method employs Kernel Mean Embedding (KME) with Isolation Kernel to represent data distributions and trains a neural network to predict the parameters of an unlearned model. The authors provide generalization bounds for logistic regression and MLP, and demonstrate efficiency improvements over existing methods on multiple datasets.

**Strengths:**

1.The idea of learning the unlearning function rather than designing it is interesting and differentiates from prior work that merely learns new decision boundaries.
2.The paper provides generalization bounds (Theorems 4.5 and 4.6) showing that the expected loss of models unlearned by L2UL is bounded close to retrained models.
3.The evaluation covers multiple datasets (7 datasets), different model types (LR, MLP, ResNet-18), and various metrics (accuracy, time, MIA, ToW).

**Weaknesses:**

1.The method requires m retrained models A(D\Df) as ground truth during training (Algorithm 1, line 7). This means the preprocessing phase involves extensive retraining - exactly what unlearning aims to avoid. If you already have the capability to retrain efficiently to generate training data, why not simply use retraining for actual unlearning? This fundamentally undermines the paper's motivation. The cost of generating m=1000 training samples with retraining (O(m·T(A(s)))) defeats the purpose.
2.Unlearning is an ongoing process (continuous forgetting) and is subjective (non-random). As unlearning progresses, the distribution of the data changes. However, the dataset D during preprocessing is fixed, and this inconsistency may impact the performance of unlearning.
3.The largest number of deletions in the experiments is 3000, which is not sufficient for large datasets like HIGGS, because deleting a small portion of data does not have a significant effect on the model.

**Questions:**

1. In line 219-222, training data  is generated by sampling D, selecting Df, and obtaining θr from A(D\Df).  This requires m retraining operations.  Given that you can retrain efficiently enough to generate 1000 training samples, why is retraining not acceptable for actual unlearning?  What is the actual cost saving in practice when preprocessing cost is considered?
2.The training uses |Df|=1 for LR and |Df|∈{1,100,1000} for MLP.  How does the learned U generalize to forget requests of significantly different sizes (e.g., trained on |Df|=100 but need to forget |Df|=5000)?  Do you need to retrain U for each forget request.
3.The main paper does not mention the hyperparameter values of L2UL, although I found them in the Appendix. I recommend including them in the main text.

---

> ### Author Response · Authors · 2025-11-20
> **rebuttal**
>
> Thank you for your review and suggestions.
>
> # Weaknesses
> A1. Instead of retraining on all the data, we sample only a small number of points for retraining. Retraining on all the data is difficult, but retraining on a small number of points is feasible.
>
> A2. Each of our retrained models is based on a sampled subset, rather than the entire dataset $D$.
>
> A3. 3000 points is small for Higgs, but our paper doesn't only use this one dataset. We also have a dataset containing 19020 points.
>
> # Questions
> A1. Retraining on a very small dataset is inexpensive, and we only need to generate such data once, rather than regenerating it every time we need to forget it.
>
> A2. In our paper, the results were obtained by fixing $D_f=10$ during training and then using the trained model to forget different $D_f=1, D_f=10$, and $D_f=1000$. We did not retrain $U$.
>
> A3. Thank you for your suggestion. Due to space limitations, we have placed it in the appendix and will move it to the main text.

---

> > ### Comment · Reviewer_pEsS · 2025-11-26
> > **Please address these specific concerns with either additional experiments or clear acknowledgment of these limitations**
> >
> > Thank you for your response. However, I still have the following concerns:
> > 1. Generalization from small-sample models: How can the unlearning function U, trained on small-sample models, generalize to models trained on the complete dataset? Your supplementary experiments in the Appendix only demonstrate that increasing s improves Accuracy, but what about the unlearning efficacy (e.g., using MIA, accuracy on D_f)?
> > 2. Contradictory statements in your response: In response A2, you mentioned "we fixed D_f=10 during training," but in the paper (footnote 4, line 289), it states: "| D_f|=1 for LR, | D_f|=1, | D_f|=100, and | D_f|=1000 for MLP." Which statement is correct? Please clarify the exact training setup used in your experiments.
> > 3. Insufficient forgetting scale: For many practical applications, forgetting 3000 out of 11 million samples (i.e., 0.03%) does not meet realistic requirements. Real-world scenarios often require forgetting a much larger portion of the dataset. Can your method handle forgetting 1% or 10% of the data? If not, this severely limits the practical applicability of L2UL.
> > Please address these specific concerns with either additional experiments or clear acknowledgment of these limitations in your paper.

---

### Meta-Review · Area_Chair_FizN · 2026-01-07

**Summary:**

The paper proposes a learning-based, model-agnostic approach to machine unlearning by learning an unlearning operator rather than explicitly designing one. Reviewers generally acknowledge that the idea is interesting and that the paper is clearly written, with theoretical analysis and empirical validation on simple models such as logistic regression and MLP.

However, reviewers raise substantial concerns regarding the methodological soundness and practical relevance of the approach. Specifically, the method relies on retraining to construct supervision for learning the unlearning operator, which weakens the motivation of avoiding retraining in the first place. Empirical evaluations are largely limited to relatively simple settings, and the effectiveness of the method under realistic large-scale models, and larger forgetting ratios remains insufficiently supported. Additionally, the reported efficiency comparisons do not include the preprocessing and training cost of the unlearning model, making the overall runtime advantage unclear.

**Reviewer Concerns:**

Concerns addressed:
The rebuttal clarifies several presentation and implementation details, including the architecture of the unlearning network U, optimizer choices, training epochs, selected hyperparameters, and missing terminology. These clarifications improve readability and reproducibility.

Concerns outstanding:
The rebuttal does not adequately address (i) the fundamental reliance on retraining to construct supervision for unlearning, which contradicts the stated motivation of avoiding retraining; (ii) the fairness of runtime comparisons, as the cost of training the unlearning model U is not included or analysed;  (iii) the scalability and practical applicability of the method to larger models, larger forgetting ratios, and realistic continuous deletion scenarios.

**Reviewer Scores:**

Reviewer pEsS: The central concern regarding the logical inconsistency introduced by reliance on retraining remains unresolved. The score would likely remain unchanged. (The reviewer actually raised further questions after reading the initial rebuttal).

Reviewer NnBA: The rebuttal does not sufficiently address scalability and cost concerns. The score would likely remain unchanged or possibly decrease slightly.

Reviewer vrxN: Concerns about the strength of theoretical guarantees, experimental rigor, and missing cost analysis are not substantively addressed. The score would likely remain unchanged.

Reviewer gR2i: Concerns about methodological complexity and circular dependence remain after rebuttal. The score would likely remain unchanged.

---

### Decision · Program_Chairs · 2026-01-26

Reject